# Deciphering Diets and Lifestyles of Prehistoric Humans through Paleoparasitology: A Review

**DOI:** 10.3390/genes14020303

**Published:** 2023-01-24

**Authors:** Rosana A. Wiscovitch-Russo, Tasha M. Santiago-Rodriguez, Gary A. Toranzos

**Affiliations:** 1J. Craig Venter Institute, Rockville, MD 20850, USA; 2Diversigen, Inc., New Brighton, MN 55112, USA; 3Environmental Microbiology Laboratory, Department of Biology, University of Puerto Rico, San Juan, PR 00925, USA

**Keywords:** ancient DNA, paleoparasitology, paleopathology, parasites

## Abstract

Parasites have affected and coevolved with humans and animals throughout history. Evidence of ancient parasitic infections, particularly, reside in archeological remains originating from different sources dating to various periods of times. The study of ancient parasites preserved in archaeological remains is known as paleoparasitology, and it initially intended to interpret migration, evolution, and dispersion patterns of ancient parasites, along with their hosts. Recently, paleoparasitology has been used to better understand dietary habits and lifestyles of ancient human societies. Paleoparasitology is increasingly being recognized as an interdisciplinary field within paleopathology that integrates areas such as palynology, archaeobotany, and zooarchaeology. Paleoparasitology also incorporates techniques such as microscopy, immunoassays, PCR, targeted sequencing, and more recently, high-throughput sequencing or shotgun metagenomics to understand ancient parasitic infections and thus interpret migration and evolution patterns, as well as dietary habits and lifestyles. The present review covers the original theories developed in the field of paleoparasitology, as well as the biology of some parasites identified in pre-Columbian cultures. Conclusions, as well as assumptions made during the discovery of the parasites in ancient samples, and how their identification may aid in better understanding part of human history, ancient diet, and lifestyles are discussed.

## 1. Introduction

Paleoparasitology, is a subdiscipline of paleopathology that includes studies of ancient parasites preserved in archeological remains [1]. The first record of ancient parasites was described by Marc Armand Ruffer, an experimental pathologist and bacteriologist who detected calcified eggs belonging to *Schistosoma haematobium* in the kidneys of 20th dynasty Egyptian mummies [2]. The field continued to develop with the identification of parasite eggs in mummified feces (coprolites) [3,4], as well as ancient sediments [5,6]. The founder and developer of the subdiscipline, Luiz Fernando Ferreira, coined the term paleoparasitology for the first time [1]. Ferreira, along with colleagues Adauto Araújo and Karl Reinhard, among others, published most of the paleoparasitology research related to pre- and post-colonization of diverse cultures throughout North and South America [1].

The founder and co-founders of paleoparasitology, who developed most known theories in the field [1], initially intended to interpret the migration, evolution, and dispersion patterns of ancient parasites and their host [7,8]. In fact, their most renowned theory is that of early human migration into the New World, where it is hypothesized that early humans crossed the Bering Land Bridge and arrived to North America about 13,000 years before present (BP) [9]. Interestingly, the founders of the subdiscipline discredit the idea of the Bering Land Bridge as being the sole migration route of pre-historic humans. To some extent, the artic environment may have potentially functioned as a barrier restricting the entrance of diseases from the Old World into the New World [9,10,11]. For instance, it is known that soil temperatures of approximately 22 °C are required for geohelminths to develop and reach an infective stage; therefore, the sub-freezing climate would have disrupted the parasites’ natural life cycles [12]. It is, therefore, theorized that a wave of early humans migrated through a transpacific route (crossing the Pacific Islands to the American Continents), allowing the proliferation and survival of the geohelminths identified today in pre-Columbian Amerindian cultures [9]. Thus, it is evident that specific host migration patterns, animal and/or human, should have facilitated the transport, dispersion and evolution of ancient parasites [9,13,14].

Human- and zoonotic parasites may have mainly resulted from specific coevolution patterns and host–parasite interactions [4]. Pre-historic hunter-gatherers were probably the most susceptible to zoonotic infections as a result of direct contact with the infected animal host and its vectors [15,16]. Although few hunter-gatherer groups have been described in paleoparasitological studies, they may have not been capable of sustaining a large parasite load, probably due to their roaming lifestyle [17]. A significant increase in parasite infection was then observed in agricultural populations, suggesting that parasite diversity increased as a result of sedentism [15,16,18,19]. By establishing permanent settlements, pre-historic humans introduced the domestication of animals as livestock and increased agricultural practices to ensure sufficient food source. This practice generated denser and larger populations that facilitated the transmission of both anthroponotic and zoonotic infections [13,14,15,17,18,20]. Overall, the level of parasitism in prehistoric agricultural villages was likely a reflection of local ecology, sanitation, behavior, and housing style of early humans.

Although paleoparasitology was initially used to determine the potential ailments of prehistoric humans, recovering parasite remnants (eggs, cysts, or larvae) from archaeological remains offers sustainable evidence of potential parasite infection(s) and insights into diets, habits and lifestyles of early humans [4,21,22]. Therefore, the present review interprets the original theories developed in the field of paleoparasitology by discussing fecal/oral, soil, and vector bone parasites identified in pre-Columbian America; the assumptions made during the discovery; and how the identification of specific parasites may aid in understanding part of human history, ancient diets, and lifestyles.

## 2. Fecal–Oral-Transmitted Parasites in Ancient Samples

The most common agents of protozoan-associated intestinal infections are *Giardia* spp. and *Cryptosporidium* spp. These parasites have the most successful rate of infection, and are commonly transmitted through fecally contaminated food, water, and fomites [23,24,25,26]. Infection with these parasites is associated with symptoms such as dehydration, nausea, vomiting, abdominal cramps, and diarrhea [24]. While giardiasis and cryptosporidiosis are common in warm and humid environments [25,27], these parasites are currently distributed worldwide and infect both humans and animals [23,24]. Notably, these diarrhea-inducing parasites are not restricted to the present era as previous studies have identified these protozoans in well-formed ancient fecal samples (coprolites) [27]. This finding is intriguing as it could suggest that early humans were susceptible to common enteric parasites, which could have caused asymptomatic or mild infections. These findings also suggest a coevolution that could have resulted from long-term, specific host–parasite interaction(s) [26].

It was long assumed that protozoan cysts could not be preserved in archeological remains since desiccation could damage the cysts over time; and in the case of the cyst being intact, it was assumed that it could be indistinguishable from fungal spores and other particles [7,26]. Thus, the identification of protozoan cysts by microscopic examination for typical morphological characteristics would have been almost impossible [7]. Other methods, such as immunoassays have also been applied to study protozoan cysts in ancient samples; yet, few paleoparasitological studies have successfully recovered protozoan cyst in ancient samples using immunoassays. Since antigens are susceptible to degradation in archeological remains, the rate of false negative results increases [28]. However, a study by Morrow et al. (2016) specifically selected coprolites that lacked the characteristic cylindrical shape, and positively identified *Cryptosporidium parvum* coproantigens in fecal samples likely resulting from diarrheal events, supporting the use of these types of tests in ancient specimens [28].

More recent paleoparasitology studies have aimed to apply molecular methods to identify specific conserved gene regions in order to have accurate taxonomical identification of ancient parasites if sufficient ancient DNA (aDNA) could be recovered [29,30,31]. While parasites, such as *Giardia* spp. and *Cryptosporidium* spp. can be host-specific or have a broad host range (Table 1) [32,33], the identification and classification of parasites in ancient samples may depend on the effect of specific taphonomical processes. Although difficult, it is not impossible to tease out or differentiate the various *Giardia* or *Cryptosporidium* spp. infecting other animals; if they were found in human coprolites, this may be a result of the ingestion of the cysts or oocysts and not necessarily a result of an active infection. Indeed, delicate protozoan cysts are infrequently detected in archeological samples compared to the more resilient helminth eggs [34]. The earliest evidence of parasite remnants was ascarid eggs dating back to the Pleistocene epoch (estimated 30,000–24,000 years-old) and were recovered from the caves of Arcy-sur-Cure located in France [35]. Similarly, nematode parasite eggs and larvae have been described in archaeological samples in the New World dating back as early as 9000 years BP [36]. This further provides evidence of the potential host migration and geographical distribution of nematode parasites [35]. Notably, certain parasites species seem to have originated in pre-hominid times and coevolved and dispersed with their host during human migration events. These parasites are referred to as “heirloom parasites”, and many were present in both humans and animals in pre-Columbian America [9].

Since most enteric parasites are currently distributed worldwide and are directly transmitted via the fecal–oral route, it is likely that enteric parasites were the most easily transmitted parasites among ancient humans. In view of the roaming lifestyle, there are few archaeological samples recognized as belonging to hunter-gatherer groups, consequentially most paleoparasitological studies have largely focused on sedentary cultures; however, there is a clear difference between the level of parasitism of hunter-gatherers and agricultural population due to potential differences in sanitation, housing style, and diets [18,19]. Initially, enteric parasites infections were less frequent in hunter-gatherers considering that small roaming groups were not stationary long enough for geohelminths to reach their infective stage [37]. Yet, during migration events of prehistoric humans, these small and diffused groups harbored and aided in the dispersion of enteric parasites and presumably other infectious diseases [9]. Enteric parasites became more frequent as prehistoric humans settled and developed agricultural practices [18,19]. Dense populations with high incidence of enteric parasites indicated poor sanitary practices by modern standards [15]. Overall, the habits of prehistoric humans facilitated the dispersion and propagation of enteric parasites. However, this may be an artifact because it may be more likely to find coprolites from sedentary cultures than from non-sedentary cultures for obvious reasons.

## 3. Soil-Transmitted Parasites in Ancient Samples

Soil has not been considered a frank source of pathogens; however, soils do play a role as possible reservoirs of certain pathogens and may act as possible vectors of certain pathogens, including parasites [38]. Soil-transmitted parasites, or geohelminths, are host-specific, are not dependent on an intermediate host, and their only known limiting factors are soil temperature and moisture [9]. The transmission and distribution of geohelminths predominantly occur throughout pantropical regions [39]. Soil-transmitted helminths, such as *Ascaris* and *Trichuris* spp., are transmitted via soil-contaminated hands or foods by ingesting the mature eggs [39]. Certain hookworks, such as ancylostomid, develop and borrow through the soil, and may penetrate the host’s skin [39]. As with the other nematode parasites mentioned above, geohelminths were present in the New World prior to European colonization and the introduction of the transatlantic slave trade [9]. Other parasites such as pinworms were also present in pre-Columbian America; however, they are not strictly regarded as a geohelminth since the parasite is not dependent on warm and humid soils for its development [9]. The transmission of pinworms such as *Enterobius* spp. is associated with direct contact and ingestion of the eggs through fomites [40]. For this reason, an outstanding question remaining is if pinworms were solely introduced into the New World through transpacific migration given that these parasites were likely capable of surviving the Beringia crossing, along with its warm-blooded host [9].

Most pre-Columbian geohelminths have been identified by microscopic analyses. For example, *Trichuris* spp. and *Ancylostomidae* are frequently detected whereas *Ascaris* spp. are rarely detected in pre-Columbian archeological samples [10,41]. A study by Leles et al. (2008) initially examined coprolites microscopically for helminth eggs [10], while the results showed the presence of *Trichuris*, *Ascaris* was not identified in the coprolite samples. This shows that microscopic examination is only successful when the sample and the parasite remnants are well-preserved [42]. Subsequently, DNA was extracted from the coprolite samples, and amplification of the *Ascaris* cytochrome b (cytb) fragment was performed. Interestingly, the results showed that four out of six coprolite samples were positive for the *Ascaris’* cytb gene. Since the composition of parasite remnants (eggs, cysts, and larvae) varies [10,26], some parasite remnants are not equally resilient to taphonomic processes [34]. These studies also show that applying a toolbox of methods for the analysis of archaeological samples could provide a better representation of the parasite composition [43]. Even though early paleoparasitological studies relied heavily on microscopy and morphology-based identification of helminths, there is the possibility that some parasite taxa may have gone unnoticed in the analysis. It should be noted that in the clinical setting, morphological identification remains as the gold standard and this is something that cannot be extended to ancient samples.

## 4. Vector-Borne Parasites in Ancient Samples

Parasitic disease vectors such as mosquitoes and triatomine bugs are responsible for the transmission of several diseases including, but not limited to leishmaniasis and trypanosomiasis. Positive identification of the parasite is important as it may indirectly determine the potential presence of the insect vector in a specific area. Insect vectors have, overall, environmental specificities, and they are prevalent in tropical regions, where warm and humid climates create a favorable condition for the development of the insect [20]; thus, the dispersion of vector-borne parasites is usually regionally limited [44,45,46]. While prehistoric human migration has influenced and facilitated the dispersion and evolution of infectious diseases, vector-borne parasites are regionally limited to certain ecological niches. Vector-borne parasites have evolved to be highly dependent on their vectors for transmission and propagation [44,45]. *Leishmania* spp. and *Trypanosoma* spp. are currently found in pantropical regions of Africa and South and Central America; and for *Leishmania* spp. this also includes some regions of the Middle East and Asia [47,48]. The insect vectors develop in warm/humid climates and reside near its warm-blooded host for direct access to blood meals [20]. As a result, these vector-borne parasites are currently limited to where their insect host can successfully reside and breed [44,45].

There is evidence of insect–parasite interactions in fossil records long before prehistoric times [49]. It is very likely that insect vectors and parasites coevolved with their vertebrate host as the insect vector fed on the vertebrate host [50]. As a result, most vector-borne diseases are primarily zoonotic, and their life cycles are highly dependent on the insect vector [44,45]. For instance, Chagas disease is endemic to Central and South America, and it is easily transmitted by triatomids, transmitting *Trypanosoma cruzi*. If the insect defecates near a skin wound, an infection then occurs as insect feces are smeared into the damaged area [47]. Trypanosomiasis, similar to other vector-borne parasite infections, is a primitive infection associated with sylvatic life cycle specific to tropical forest environment. Notably, *T. cruzi* does not seem to affect the insect vector itself, suggesting a long period of parasite/insect vector adaptation and coevolution [51]. The trypanosome insect vector likely adapted to human habitations for easy access to blood meals [51]. The triatomid vector resides near its blood meal, and can hide in cracks and straw roofs of adobe houses, and feeds off the inhabitants and warm-blooded animals (e.g., camelids, dogs, and rodents) living near human housing area [29,52,53]. Thus, it is unlikely that Chagas disease developed during the nomadic hunter and gatherer stages of early humans, and most likely became endemic in the Andean region after the establishment of permanent settlements and the adaptation of the triatomid vector to human dwellings [53,54].

Indeed, sedentarism in prehistoric humans initiated the early development of agricultural practice and domestication of wild animals to ensure a sufficient food stock. Essentially, sedentary habits stimulated the adaptation of triatomids to human dwellings by keeping animals as pets or livestock [54]. In addition, grain storage likely attracted wild grain feeding mammals (e.g., rodents), facilitating the arrival of triatomids and other parasite-carrying insects (e.g., grain beetle to human dwellings [54]). Historically, some rodent species lived near human settlements [55], providing food, shelter, and protection against other small rodents from predatory species. In addition, canids were also regarded as potential reservoirs of leishmaniasis, as canids were considered both pets and an occasional protein food source in prehistoric human settlements [56,57].

The first evidence of trypanosomiasis and leishmaniasis was excavated from pre-Columbian burial sites [52]. Initially, the *Trypanosomatidae* family, which includes the genera *Leishmania* and *Trypanosoma*, was difficult to identify in pre-Columbian samples [52]. Archeological artifacts such as small clay burial statues (known as huacos) represented individuals with facial deformities (i.e., nose, eyes, and mouth lesions), similar to mucocutaneous leishmaniasis symptoms [52,58,59]. In addition, paleopathology examination of skeletal and mummified human remains identified symptoms similar to leishmaniasis and trypanosomiasis (e.g., dilated heart, esophagus, and/or colon). DNA-based methods have been applied to identify *Leishmania* spp. and *Trypanosoma* spp. genomic regions. For instance, Costa et al. (2009) and Marsteller et al. (2011) examined skulls with evidence of severe destruction in the oral-nasal and pharyngeal cavities associated to chronic leishmaniasis [58,59]. Both studies confirmed the diagnosis of leishmaniasis by amplifying conserved gene regions of the pathogen, further confirming that the facial deformities were caused by *Leishmania* spp. and were not associated with other facial deformities related to cancer, leprosy, trypanosomiasis, or tuberculosis [56,58,59]. The *Trypanosomatidae* family has also been identified in insect vectors (e.g., *Phlebotomidae* and *Triatominae*) preserved in Dominican amber (estimated 20–30 million years ago) [50,60]. Specifically, the malaria parasite (*Plasmodium* spp.) has been identified in a mosquito vector preserved in Tertiary Dominican Amber [61].

Target-based and shotgun metagenomic sequencing can potentially also provide accurate species-level identification of vector-borne parasites, and further interpretation of the disease. For instance, Lima et al. (2008) and Fernandes et al. (2008) used target-based sequencing to identify *Trypanosoma cruzi* I in human remains [30,54]. The genotype has a wide host range predominantly associated to sylvatic transmission cycle and is mainly associated to human disease endemic to the Amazonian Basin [45,54]. Using high-throughput next generation sequencing and metagenomic methods, *T. cruzi* (homologous to strains CL Brener and Esmeraldo) and *Leishmania donovani* were identified in the descending colon of a pre-Columbian Andean mummy [31]. Although shotgun metagenomic sequencing typically produces large metagenomic datasets, the damage inflicted on the DNA by the taphonomic processes may not enable the reconstruction of the whole genome of ancient parasites [62]. In addition, comparative genomic analyses between ancient and modern parasites may provide further perspective into the evolution of these pathogens [31].

## 5. Zoonotic Tapeworms in Ancient Samples

Tapeworms are zoonotic parasites that can be transmitted to humans through an intermediate host and are known to have affected prehistoric humans. As mentioned, identification of ancient parasites with intermediate host allows the inference of prehistoric human’s diet and habits [21]. While the preservation of parasite eggs is essential for accurate morphological-based identification, species-level resolution is needed to determine the precise host-range of a parasite. In the case of cestodes or tapeworms, similar morphological features are shared within the class, further complicating microscopic examination for species classification [34]. In addition, taphonomical damage inflicted on the parasite egg may hinder its identification [34,37]. As a result, few paleoparasitological studies have confidently identified cestodes eggs by microscopic examination and were only capable of providing a generic identification of the parasite. Zoonotic tapeworms have complex life cycles infecting multiple hosts through several modes of transmission (Figure 1). Figure 1 shows examples of zoonotic tapeworms indirectly infecting their human host after ingesting an infected intermediate host. The network modeling was generated as described previously [46], and shows previous studies by Santoro (2003) (Figure 1A) [63], Patrucco (1983) (Figure 1B) [21], Reinhard (1987) (Figure 1C) [15], and Jimenez (2012) (Figure 1D) [64]. As seen in Figure 1, the represented ancient human populations were mostly susceptible to parasite transmitted via fecal-oral route (e.g., *Ascaris, Trichuris* and *Enterobius* spp.) or through ingestion of infected intermediate host (e.g., *Diphyllobothrium, Dipylidium* and *Hymenolepis* spp.). See [46] for more information.

As mentioned, the selected studies shown in Figure 1 represent examples of parasites infecting humans and animals and potential modes of transmission. For instance, the study by Santoro et al. (2003) investigated the helminthological composition of coprolites recovered from Lluta Valley (Chile) dating to both the pre-Inca and Inca periods [63]. Lluta Valley is characterized by a variety of aquatic environments including freshwaters, marine waters, and estuaries that promoted subsistence and commerce in ancient cultures. Pre-Inca settlements in this region consisted of a small community with residences and at least one cemetery, whereas the Inca settlements were larger in size, and with public architecture. Notably, coprolite analysis also showed that the diet of pre-Inca cultures consisted of, for the most part, local food items, whereas the diet of Inca cultures in this region consisted of food items of both local regions, as well as those obtained through trade [63]. As shown in Figure 1A, inhabitants from the Lluta Valley potentially hosted multiple species of parasites with diverse modes of infection (e.g., fecal-borne), as well as associated those with diet (e.g., poorly cooked fish), person to person contact and contact with contaminated fomites [63].

Pre-Columbian cultures in the North American Southwest have also been investigated through paleoparasitological analysis of coprolites, as shown in Figure 1C [15]. The study by Reinhard et al. (1987) investigated over 300 coprolite samples from six different sites, three of which were caves, of both prehistoric hunter-gatherers and agriculturalists. One of the caves, known as Dust Devil Cave, was occasionally used by a small group of nomadic hunter-gatherers about 8000 to 6000 years ago. Interestingly, coprolites from this site showed no evidence of helminth remains. Another cave, known as Turkey Pen Cave, was used by a group of agriculturalists approximately 1600 years ago, which also foraged for wild plants. A third cave, known as Antelope House, was used by corn agriculturalists from 200 A.D. to 1250 A.D. Notably, several of the coprolites found in this cave belonged to dogs, suggesting that dogs were one potential reservoir and potentially responsible for the transmission of parasites (Figure 1C). According to the study, coprolites from Antelope House were distinct from the other agricultural sites in the number of helminths species identified. Two of these helminth species—namely, *Strongyloides* sp. and *Trichostrongylus* sp.—are known to be dependent on moist soils for the completion of their life cycles, indicating that foraging in moist areas exposed the inhabitants to these parasites. These parasites are probably associated to diet, and the utilization of specific food items [15].

A follow-up study connecting parasitological observations between the North American Southwest and Mesoamerica was performed by Jimenez et al. (2012) (Figure 1D) [64]. The archeological site studied was Cueva de los Muertos Chiquitos, located in the Northern Durango region of el Zape, representing a transition zone between the North American Southwest and Mesoamerica. Among the identified parasites included six species; three of which possess a monoxenous (i.e., development depends on a single host species) and three a heteroxenous (i.e., development depends on at least two types of hosts) life cycle. All the heteroxenous parasite taxa identified in the study were mostly associated with rodents and dogs, with humans serving as occasional hosts. As mentioned, the presence of parasites of rodents in ancient human feces has been associated with storing agricultural goods in granaries, which attract arthropods and rodents [64]. Transmission from dogs to humans may be associated with the consumption of fleas or lice infected with *Dipylidium caninum*, which use fleas as intermediate hosts [64].

Zoonotic tapeworms can be associated with ingestion of an infected animal. For instance, *Diphyllobothrium* spp. (reclassified as *Dibothriocephalus* spp.) is a fish-borne tapeworm that affects fish-eating mammals acquired after ingesting raw or undercooked fish [65], and eggs have been recovered from both human and canid coprolites [22,65]. In pre-Columbian cultures, diphyllobothriasis was more common in coastal fishing populations; thus, a low incidence of diphyllobothriasis was detected in certain inland populations [20,66]. Inland agriculturalists were exposed to diphyllobothriasis by the trade of food items and other goods between coastal and inland populations [20,66]. Nevertheless, canids were also susceptible to diphyllobothriasis and would potentially get infected by consuming food scraps of infected fish. In general, humans keeping animals as pets or livestock would potentially expose them to zoonotic tapeworms [67]. For instance, dogs, specifically, were highly revered in the agrarian and pastoral Chiribaya society, as suggested by the mummified canid buried alongside the human owner’s corpse [65]. Close contact with infected canids most likely made the Chiribayan susceptible to the nematode *Toxocara canis* and most certainly other forms of zoonotic infections, such as the canid tapeworm *Dipylidium caninum* [65].

Zoonotic tapeworms are also associated with direct contact with infected animals and have also been identified in archeological records. For instance, zoonotic parasites *D. caninum* and *Hymenolepis* spp. eggs were detected in 1400-year-old coprolites of Cueva de los Muertos Chiquitos, Mexico [64]. Indeed, canids are known to be a reservoir of *D. caninum*. Currently, human infections of the double-pored tapeworm are rare, but it is associated with having close contact with flea-infected pets [68]. In addition, human infection is related with accidental ingestion of the cysticercoid contaminated flea vector [68]. Prehistoric cultures were known to control lice infections by ingesting the lice while grooming [34]; thus, eating the infected flea vector could have been used to prevent the ectoparasite from feeding off the human or canid host [20]. Fugassa et al. (2011) identified masticated tick remains in human coprolites from Antelope Cave; thus, it is evident that the inhabitant of the cave ingested the ticks [67]. Undoubtedly, some prehistoric cultures ingested insect vector to control an outbreak or were simply consumed as a protein food source. Another example of a zoonotic tapeworm infection associated with ingestion of an insect vector is hymenolepiasis. *Hymenolepididae* human infections are associated with contaminated grain storage [20]. Regarding the parasite’s life cycle, *Hymenolepis* spp. is mainly related to agricultural groups [15,64]. Theoretically, *Hymenolepididae* infection would be more prevalent in agricultural groups due to their habit of storing the surplus grains [20]. This habit would have attracted grain feeding insects and rodents to the dwellings and potentially sustain *Hymenolepis* infection [20]. While Hymenolepidids commonly infects rodents, *H. nana* can also infect humans as definitive host [69]. Due to the grain beetle’s size (2–3 mm), it is possible that prehistoric humans did not bother removing the beetle before ingesting the contaminated grain [19,20]. Overall, it is apparent that identifying dipylidiasis and hymenolepiasis in human coprolites suggests that prehistoric humans ingested the intermediate host and had close contact with infected animals [21].

Description of the parasites mentioned throughout the above sections are summarized in Table 2. Table includes examples of parasites detected, country or region, type of archeological samples, estimated date, and the method of detection. As mentioned, these parasites are transmitted through, but not limited to the fecal–oral route, soil, and other vectors (see text).

## 6. False Parasitism in Ancient Samples

False parasitism is defined as a parasite recovered from an unusual host. Discovering a non-human parasite from human fecal samples would be an example of false parasitism. However, the finding of a false parasite in coprolites may also provide evidence of ancient human’s diets [19]. For instance, *Eimeria* cysts have mainly been identified in pre-Columbian camelid coprolites and mummified tissue [70,71,72] and have occasionally been recovered from atypical host coprolites [73]. *Eimeria* spp. is of veterinary and economic importance in the livestock industry [74], as it is a parasitic disease of the intestinal tract of animals, particularly domesticated birds (fowls) [75]. However, in South America, coccidiosis, which is caused by *Eimeria* spp., is a common infection in native camelids such as llamas, alpacas, vicuñas, and guanacos [74]. Specifically, *Eimeria macusaniensis* is one of the five most prevalent species of *Eimeria* in South American camelids [74], and cysts have been identified in felid coprolites. *Eimeria mancusaniensis* was most likely acquired by a big feline (potentially *Puma concolor*) after consuming parts of the viscera of an infected *Camelidae* [73]. In addition, *Eimeria* sequences have been identified in metagenomic dataset produced from pre-Columbian human coprolites, and a variety of fowl osseous remains were described in the zooarchaeological data [76], thus suggesting the consumption of raw or undercooked infected birds [46]. Depending on the species of *Eimeriidae*, the parasite is highly host-specific [75], and the ingestion of *Eimeria* infected tissue would have never emerged within an atypical host, but rather, it would have become a transient organism.

## 7. Paleoparasitology as an Interdisciplinary Field

As highlighted in the present review, paleoparasitology is an increasing developing field whose main purpose is to, not only study ancient cultures but also find further interpretations. Notably, paleoparasitology is increasingly employing a number of disciplines including, but not limited to palynology, archaeobotany, and zooarchaeology to further its purpose [77]. An example of this is the consideration of faunal osseous remains to provide the potential of transmission of zoonotic infections [64]. Faunal osseous remains extracted from archaeological deposits do not only disclose the local fauna surrounding the human habitat [78] but also provide insights into ancient human’s potential protein food source(s) [46,79]. Moreover, archaeological samples (soil, feces, or dental calculus) can be subjected to microscopic examination for the detection of microfossil such as pollen grains, spores, phytoliths, starch granules, and other debris. These preserved microfossils can reveal potential plant-based diets and horticultural preferences of ancient cultures [77,78,79]. Furthermore, the examination of mummified feces (extracted from the abdominal cavity of a mummy) can reveal what an individual consumed shortly before death [80]. For instance, Allison et al. (1974) identified intestinal content of a Tiahuanaco mummy that was consistent with corn, beans, meat (likely charque/jerky), and several other vegetables [80]. Additionally, feathers, hair, and small animal bones have been recovered from ancient feces and are also potentially an indicator of protein sources [37,66]. This discovery suggests that the animals were potentially too small to remove the bones or the integuments (i.e., feathers and hair) from the meat, and thus, were ingested whole [66]. Indeed, these studies indicate that the incorporation of different knowledge and fields can be used to reconstruct the diets of ancient humans.

Recently, molecular methods (amplicon or shotgun metagenome sequencing) have been used to infer the diets of ancient humans by assessing homology reads associated to animal or plant proteins. However, taphonomic processes cause aDNA damage, which results in low-quality alignments between extant and ancient DNA sequences. For this reason, palynology, archaeobotany, and/or zooarchaeology analyses of the sample need to complement sequencing information [46,79]. While archaeological samples are a wealth of information, extensive training and collaboration in a multidisciplinary scientific analysis are usually required [77].

## 8. Inferring Lifestyles and Diets of Ancient Cultures from Extant Isolated Cultures

The present review highlighted the potential of paleoparasitology to infer lifestyles and diets of ancient cultures. However, the disease ecologies of extant indigenous groups living in remote areas are also a suitable model for comparing and predicting the behavior of ancient humans [81]. For instance, the Yanomami culture are a semi-isolated indigenous group currently residing in the Amazonian jungle of Venezuela [82]. Although the Yanomami have established permanent settlements, they are best described as a hunter-gatherer culture [83]. It is known that these cultures are susceptible to geohelminths and intestinal protozoan parasites due to the cultures feeding habits and constant contact with feces-contaminated soil [83]. However, the parasite burden varies between sedentary and semi-nomadic populations [84]. Two indigenous populations residing in the Amazonian jungle, the Tukano and the Maku [84]. Intestinal protozoan parasites and geohelminths were present in both populations, but the Tukano had a lower incident of geohelminths. The Maku culture roaming lifestyle does not have a prolonged contact with fecally contaminated soil and could potentially limit the exposure to geohelminths [84]. The presence of geohelminths in large and dense sedentary populations reflected the poor sanitation within the inhabiting areas (the dwellings or agricultural fields) by modern standards [20,41,83,84]. Geohelminths tend to live in close proximity to their host, thus human settlements facilitate the transmission of these pathogens [85]. Overall, human settlements possibly seem to have resulted in sustainable and reoccurring parasite infections [13,15,16,17,86].

## 9. Conclusions

Examining parasite remnants in ancient samples provides further understanding of prehistoric humans. Paleoparasitology does not only reflect the diets and cultural habits but also allows for the interpretation of migration patterns, occupation, trade, sanitation, domestication of animals, and agricultural practices of ancient cultures. Essentially, human migration and trade facilitated the dispersion of ancient parasites; however, parasite infections were successful only when the conditions were favorable for the parasites to complete their life cycles. As mentioned, this is mostly dependent on environmental factors (temperature and moisture) and vector presence in the new environment. Hence, fecal–oral parasites were probably the most easily transmitted and dispersed parasite in early humans. As ancient humans transitioned from hunter-gatherer to agriculturalist, depending on the population density and stationary time, determined the parasite diversity and frequency of infection. Consequently, large human settlements exhibited greater parasite diversity and infections, both from fecal–oral transmitted and zoonotic parasites. In dense populations, detecting enteric parasites in prehistoric dwellings reflected poor sanitation by modern standards and animals (domesticated or feral) present in human settlements were likely reservoirs of zoonotic parasites. As paleoparasitology moves forward as a field, it is anticipated that it will continue to be interdisciplinary, incorporating various fields including, but not limited to, palynology, archaeobotany, and zooarchaeology and use a variety of techniques such as microscopy, immunoassays, molecular methods, and high-throughput or shotgun metagenomic sequencing. The incorporation of multiple fields and techniques in paleoparasitology will continue to provide insights into ancient dietary habits and lifestyles that are an intrinsic part of human history.

## Figures and Tables

**Figure 1 genes-14-00303-f001:**
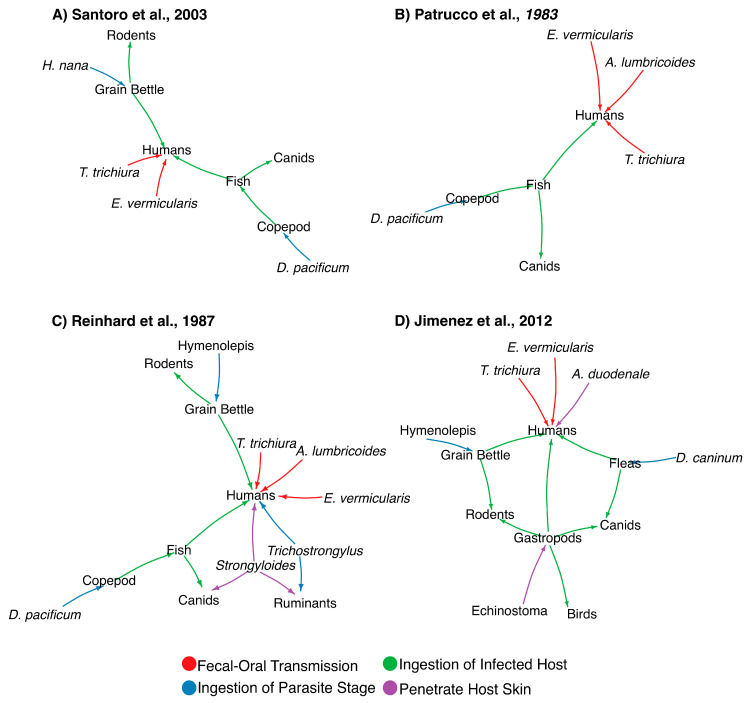
Network modeling representing the parasite-host interactions of pre-Columbian cultures. Networks were generated from previous studies as described by Santoro (2003) (**Panel A**) [63], Patrucco (1983) (**Panel B**) [21], Reinhard (1987) (**Panel C**) [15], and Jimenez (2012) [64] (**Panel D**) (see text). Figure was modified from [46]. Original figure was published under a Creative Commons Attribution License, which permits unrestricted use, distribution, and reproduction in any medium, provided the original author and source are credited.

**Table 1 genes-14-00303-t001:** *Cryptosporidium* spp. and *Giardia* spp. potential host range. Information from [32,33].

Genus	Species	Potential Host Range
*Cryptosporidium*	*C. hominis*	Humans
	*C. parvum*	Mammals (humans, ruminants, rodents)
	*C. muris*	Mammals (humans *, ruminants, rodents)
	*C. canis*	Dogs, humans*
	*C. meleagridis*	Birds and humans
	*C. baileyi*	Gallinaceous birds
	*C. galli*	Birds
	*C. molnari*	Fish
	*C. nasorum*	Fish
	*C. wrairi*	Guinea pigs and humans *
*Giardia*	*G. duodenalis, G. intestinalis,* and *G. lamblia*	Mammals (humans, rodents, canids, ruminants), birds, and reptiles
	*G. muris*	Rodents
	*G. microti*	Rodents
	*G. agilis* and *G. gracilis*	Amphibia, birds, reptiles
	*G. psittaci*	Birds
	*G. ardae*	Birds

* Rare infections.

**Table 2 genes-14-00303-t002:** Description of parasites identified in prior paleoparasitological studies. Table includes examples of parasites detected, country or region, type of archeological samples, estimated date, and the method of detection.

Example Parasite Detected	Region or Country	Archeological Sample	Estimated Date	Method of Detection	Citation
*Diphyllobothrium pacificum/* *Enterobius vermicular/* *Ascaris*	Peru	Human Coprolites	4800–3750 B.P.	DM	[21]
*Trypanosoma*	Chile	Human Mummified Tissue	1600–2420 B.P.	Autopsy	[53]
*Enterobius vermicularis/* *Trichostrongylus* sp./ *Strongyloides* sp.	USA	Coprolites/Soil	1000–8000 B.P.	DM	[15]
*Cryptosporidium/* *Giardia*	Andenian Region	Human Mummified Feces	500–3000 B.P.	FM	[27]
*Giardia duodenalis*	USA	Unidentified Coprolite/Sediment	1200–1300 A.D.	ELISA	[7]
*Enterobius vermicularis/* *Trichuris trichiura*	Chile	Human Coprolites	1200–1500 A.D.	DM	[63]
*Enterobius vermicularis*	Chile and USA	Human Coprolites	4110 B.C.–900 A.D.	DM/PCR	[42]
*Trypanosoma cruzi I*	Brazil	Human Bones	7000–4500 B.P.	TBS	[30]
*Trypanosoma cruzi I*	Brazil	Human Mummified Tissue	560 ± 40 B.P.	TBS	[54]
*Ascaris*	Brazil and Chile	Human Coprolites	8800–430 B.P.	DM/PCR	[10]
*Leishmania*	Chile	Human Skeletal Remains	500–1000 A.D.	PP/PCR	[58,59]
*Eimeria macusaniensis/* *Calodium* spp.	Argentina	Unidentified Coprolites	3480–2740 B.P.	DM	[37]
*Enterobius vermicular/* *Trichuris vulpi/Acanthocephala*	USA	Human/Canidae Coprolites	680–960 A.D.	DM	[67]
*Plasmodium vivax*	Peru	Human Mummified Bodies	3000–600 B.P.	ELISA	[52]
*Enterobius vermicularis/* *Echinostoma/* *Hymenolepis*	Mexico	Unidentified Coprolites	1400 B.P.	DM	[64]
*Diphyllobothrium/* *Toxocara canis/* *Trichuris vulpis*	Peru	Canidae Coprolites	700–1476 A.D.	DM	[65]
*Leishmania donovani/* *Trypanozoma cruzi*	Peru	Human Mummified Tissue	980–1170 A.D.	HTS	[31]
*Cryptosporidium parvum*	Mexico	Unidentified Coprolites	1200–1400 B.P.	ELISA	[28]
*Diphyllobothrium/* *Dipylidium caninum/* *Cryptosporidium spp./* *Giardia intestinalis/* *Schistosoma* spp.	Puerto Rico	Human Coprolites	215–600 A.D.	DM/HTS	[46]

SEM = Scanning Electron Microscopy; DM = Direct Microscopy; IH = Immunohistochemical; EM = Electron Microscopy; FM = Fluorescent Microscopy; TBS = Target-Based Sequencing; PP = Paleopathology; HTS = High-throughput sequencing. B.P. = Before Present. A.D. = Anno Domini.

## Data Availability

Not applicable.

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
