# Peer review of "Deciphering Diets and Lifestyles of Prehistoric Humans through Paleoparasitology: A Review"

_genes, 2023, doi:10.3390/genes14020303_

Round 1
Reviewer 1 Report
The authors have presented a comprehensive review explaining prehistoric diets and lifestyles through paleoparasitology with an emphasis on parasitic interactions between the hosts and vectors. No obvious weaknesses or flaws. This study will only add to the existing literature. The conclusions are well supported by the evidence presented.
Author Response
We thank the reviewer for revising our manuscript.
Reviewer 2 Report
It is an interesting paper which aims at synthetizing data concerning ancient parasites identified in archeological remains found in the American continent and infering some informations concerning the ancient diet and lifestyles. Some additional elements would be necessary :
1/ Table 1: the datation of the different samples is lacking
2/ For the figure 1, it will be interesting to specify what type of site is concerned and their characteristics concerning the fauna and the environmental conditions
For vector-borne diseases, there are specific biotopes conducive to the proliferation of vectors. That is why it would be interesting to have such informations!
It would be important to cite : Carvalho Gonçalves, Araujo, Ferreira 2003 Mem. Inst. Oswaldo Cruz 98
Author Response
We thank the reviewer for revising our manuscript.
1/ Table 1: the datation of the different samples is lacking
We thank the reviewer for this suggestion. We have added the datation to Table 1 (new Table 2).
2/ For the figure 1, it will be interesting to specify what type of site is concerned and their characteristics concerning the fauna and the environmental conditions
For vector-borne diseases, there are specific biotopes conducive to the proliferation of vectors. That is why it would be interesting to have such informations!
We thank the reviewer for these suggestions. We have now added more information regarding type of site and several of the characteristics related to fauna and environmental conditions to the revised version of the manuscript (lines 292-334). Concerning the biotopes, the information can be found throughout the manuscript text.
It would be important to cite : Carvalho Gonçalves, Araujo, Ferreira 2003 Mem. Inst. Oswaldo Cruz 98
We thank the reviewer for pointing to this reference. We have now included this reference as new reference 11.